# CoLLIE: Continual Learning of Language Grounding from Language-Image Embeddings

## Abstract

This paper presents CoLLIE: a simple, yet effective model for continual learning of how language is grounded in vision. Given a pre-trained multimodal embedding model, where language and images are projected in the same semantic space (in this case CLIP by OpenAI), CoLLIE learns a transformation function that adjusts the language embeddings when needed to accommodate new language use. Unlike traditional few-shot learning, the model does not just learn new classes and labels, but can also generalize to similar language use. We verify the model's performance on two different tasks of continual learning and show that it can efficiently learn and generalize from only a few examples, with little interference with the model's original zero-shot performance.

## 1 Introduction

Any artificial agent interacting with an environment, using vision, and communicating with other agents (such as humans), using language, needs to be able to ground the meaning of language with the visual properties of the environment. One approach to this problem is to project vision and language into a joint semantic embedding space (Frome et al., 2013; Bruni et al., 2014). In such a model, a visual stimulus and a language construct that have similar representations are supposed to have similar meanings. In order to name a given object with certain visual features, the agent should try to generate a referring expression that has a similar embedding as the visual features of the object, and in order to understand what a referring expression is denoting, it should look for objects that have a similar visual feature embedding as that of the referring expression.

Recent developments in multimodal representation learning using large amounts of data have given impressive results. An example of a model integrating language and vision is CLIP by OpenAI (Radford et al., 2021), which was trained using constrastive learning on 400 million pairs of images and their captions. Images and texts are embedded (separately) using state-of-the-art computer vision and language processing pipelines into a 512 dimensional vector. By calculating the dot product of the two embeddings, it is possible to determine how similar an image is to a text (or an image to an image, or a text to a text), as illustrated in Figure 1a. The model was shown to be very effective at so-called *zero-shot learning*, which for CLIP means that the model can do image retrieval by ranking the similarity of images to a given label (such as "a black cat"). This can be contrasted with traditional image classification, where the model is specifically trained to classify images into a predefined set of categories (e.g., Deng et al. 2009). In addition to being more flexible (as it can use a virtually infinite set of categories), CLIP was also shown to be more robust against noise and variations in the images, compared to supervised image classification (Radford et al., 2021).

While such a model is potentially very useful for agents that need to ground language in vision, it is limited in that it is trained once, without any mechanism for updating its representations in light of new data, unless the entire model is retrained and the number of new examples is sufficient. This is clearly limiting the model's usefulness in real-life application scenarios for agents interacting in a dynamic environment. Not only will new objects with new properties emerge, but the way humans talk about objects changes over time. As has been shown repeatedly in experiments on human-human interaction, this is not only a long-term issue, but the exact meaning of language may often be negotiated and evolve during the course of a single interaction, and then develop into partner-specific language use (Brennan & Clark, 1996; Shore & Skantze, 2018). This phenomenon has been referred to as *conceptual pacts* (Brennan & Clark, 1996), or more generally as *alignment*

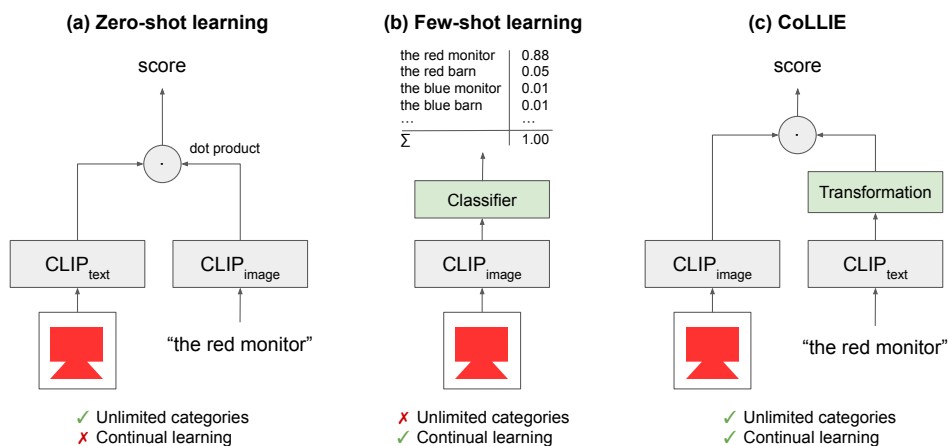

Figure 1: Comparison of CoLLIE to Zero-shot and Few-shot learning. Green boxes show where continual learning is taking place.

in communication (Pickering & Garrod, 2006). For example, if a hard-to-describe object is being referred to, the partners might establish a new name for it and then continue using that name for similar objects.

The ability to continually learn over time has been a long-standing challenge for machine learning and artificial intelligence, and this area of research has been referred to as *continual* or *lifelong learning* (Parisi et al., 2019). There are several problems involved in this. First, humans can learn new concepts using only a few examples, in contrast to machine learning models that typically need several orders of magnitude more examples. Second, computational models have been shown to be prone to so-called *catastrophic forgetting* (Parisi et al., 2019). Unless the model is re-trained entirely from scratch (which is infeasible for large models like CLIP), the updates to the model's parameters might interfere with previously learned knowledge, resulting in abrupt performance drops. This is also referred to as the *stability-plasticity dilemma* (Parisi et al., 2019).

Learning from few examples greatly depends on having powerful enough representations. Thus, the first problem has been addressed using *transfer learning*, where a fixed *base model* learns rich general representations from other (but related) tasks. This base model is then used as input to a simple classifier with only a few parameters (such as logistic regression), requiring only a few training examples. This is often referred to as *few-shot learning* (Wang et al., 2020). Since the CLIP model is trained to learn powerful general representations, it was also shown to be fairly good as a base model for few-shot learning (Radford et al., 2021). In principle, an agent that sees a new object and hears it being referred to as a "a red monitor" by a human could train a few-shot classifier to be able to identify such objects in the future, as illustrated in Figure 1b. However, a problem with this form of few-shot learning is that it is based on the same principles as the conventional supervised image classification discussed above, where labels do not have any inherent meaning, but are instead treated as atomic symbols. If an agent using a language-image embedding model (such as CLIP) would learn a new label using this approach, it is not clear when it should use its base model to resolve language-image relationships (as in Figure 1a), and when it should apply the newly learned classifier for the specific label (as in Figure 1b). Moreover, those new categories would have no relationship to other (previously known or newly acquired) categories. For example, if the agent would learn how the term "the red monitor" is used in a specific situation, it would not be able to infer that "the red display" might be used in a similar way. In addition, it is unclear how it should be able to make use of *semantic compositionality*, i.e., to combine (in a principled way) the newly acquired language with the language it already knows in a compositional manner, to understand expressions such as "the blue monitor".

In this paper, we propose **CoLLIE**, a simple, yet effective, model for **Co**ntinual learning of **L**anguage grounding from **L**anguage-**I**mage **E**mbeddings. The general principle of CoLLIE is illustrated in Figure 1c. Instead of learning a new model for each new concept (as in few-shot learning),

the model relies on a base model of language-image embeddings in a joint embedding space (CLIP in our case), with zero-shot capabilities. We then use and update a separate *transformation model* that makes adjustments to the language embedding to better fit the new concepts that are being learned, when needed. Thus, the continual learning is only taking place in this transformation model, while the base model is fixed. Our aim is to achieve the following characteristics:

- **Sample efficient**: We want to be able to learn new language-image mappings quickly with only a few examples.
- **Computationally efficient**: The transformation model is very lightweight and relatively cheap to retrain. While ideally continual learning should happen without any retraining (so-called "rehearsal") and without keeping training data in memory (Parisi et al., 2019), we accept rehearsal given that only the transformation model needs to be re-trained. We will return to this issue in the discussion section.
- **Generalizable**: We want to be able to use the newly learned concepts to understand new related concepts.
- **Robust**: As the model learns new concepts, it should continue to perform equally well on tasks it could do before, and newly learned concepts should not interfere with each other.

## 2 RELATED WORK

Language grounding is a core problem of AI, and is related to the more general problem of symbol grounding. i.e., how the symbols used by an AI system get their meaning in terms of how they are anchored to the external world (Harnad, 1990). In the field of computational linguistics, there is a long history of research on how to infer the referents of referring expressions (so-called exophoric reference resolution), as well as how to generate referring expressions, based on the visual properties of the target referent and potential distractors (Krahmer & van Deemter, 2012; Qiao et al., 2020). In the field of computer vision, the related problems of image captioning (generating a text describing an image) and image retrieval and object detection (from natural language descriptions) have also been studied extensively (e.g., You et al. 2016; Hu et al. 2016). Recently, there has also been a lot of research done in the areas of Visual Question-Answering (VQA) and Visual Dialog, where visual language understanding and generation is combined and treated in an end-to-end fashion (Kafle & Kanan, 2017; Das et al., 2017). However, these studies typically assume that a fixed model of language grounding can be trained, and that the language use does not change after that.

Previous research on continual learning in image classification has mainly studied the effect of adding new classes to the model (Kemker & Kanan, 2018; Kemker et al., 2018). Our work is different, in that it is not based on image classification with a limited set of classes, but rather on adjusting a model that can do zero-shot image retrieval, where the language can form a virtually endless number of "classes".

The challenge of how to achieve continual learning without catastrophic forgetting has been studied for a long time, since early studies of continual learning in neural networks showed that this was indeed a serious problem (McCloskey & Cohen, 1989). Parisi et al. (2019) outline three basic approaches to alleviate catastrophic forgetting for continual learning in neural networks: First, various regularization approaches may be used to impose constraints on the update of the model's parameters. Second, it is possible to allow the architecture of the network to change, e.g., by adding neurons or layers. Third, complementary learning systems are inspired by the human brain, in that they rely on an interplay between episodic memory (specific experience) and a semantic memory (general structured knowledge), where learning first happens in the former, and is eventually consolidated with the latter (during "sleep"). CoLLIE does not fit squarely into any of these, but comes closest to complementary learning in its use of a base model (where parameters are fixed) and a dynamic model (where learning happens).

## 3 DATA, TASK AND METRIC

For our evaluations, we assume that the task is to rank a set of candidate referents based on how well they match a referring expression. As our metric, we use the Mean Reciprocal Rank (MRR),

which is equal to 1 divided by the assigned rank of the correct candidate, yielding a score between 0 and 1. Thus, an MRR of 1 corresponds to ranking the correct candidate first and 0.5 corresponds to ranking it second (which can still be considered quite good if the number of candidates is large). The reason we choose MRR instead of accuracy is that it does not only take the top-ranked candidate into account, and therefore can be considered to be a more nuanced metric.

In this paper, we use two datasets. First, we use the **LAD dataset** (Large-scale Attribute Dataset) by Zhao et al. (2018), from which we selected a set of 200 categories belonging to the super-categories animals, fruits, electronics and vehicles, with a total of 68247 images. To verify CLIP's zero-shot performance on this dataset[1], we did 20 iterations where we randomly selected one image per category (i.e., 200 images) and performed the ranking task using the LAD labels of the categories as referring expressions, yielding an MRR of 0.773. We think this confirms CLIP's impressive zero-shot performance on these types of images.

To study a more challenging set of images (for CLIP), we also use the images from the **KTH Tangrams dataset** (Shore et al., 2018). Orginially, these images were used in an experiment where participants were asked to take turns referring to tangram figures (using spoken language) while the other participant was trying to identify them on a common game board (Shore & Skantze, 2018). To assess CLIP's zero-shot performance on these tangram figures, we took the 17 shapes used in the study and made colored versions of them (red, green, blue, yellow and purple), constituting a set of 85 candidate referents. The referring expressions were constructed by combining the color with the name of the shape used by the authors of the paper (e.g., "the blue giraffe"). As expected, CLIP's zero-shot performance on these referents is not as good, only yielding an overall MRR of 0.31. The MRR for the individual shapes are shown in the Appendix (Figure 7). While some shapes are identified correctly ("mountain", "barn"), most of them are not. This is of course understandable, given that these images are not representative of CLIP's training data.

In fact, it was not that easy for the human participants in the experiment to do this task either, at least not for their initial attempts. However, they soon started to invent names for the different shapes, forming conceptual pacts after repeated interactions and making the interactions more efficient over time. If an artificial agent should be able to engage in such a task, it would clearly have to be able to apply some form of continual learning in the way outlined in the introduction. For this to work, CLIP still needs to have a good representation of the images. To investigate whether this is the case, we performed a t-SNE analysis (Van Der Maaten & Hinton, 2008) on the CLIP embeddings of the colored tangram images to reduce the 512 dimensions to 2 dimensions, as illustrated in Figure 6 in the Appendix. As can be seen, the shapes and colors seem to form clusters and to be handled in a somewhat consistent fashion, which indicates that it should indeed be possible to learn names for them.

## 4 THE CoLLIE TRANSFORMATION

The idea behind CoLLIE is to learn a **transformation function**, $\tau = T(t) : \mathbb{R}^{512} \to \mathbb{R}^{512}$, which takes the CLIP embedding of the text, $t$, and returns another transformed embedding, $\tau$, that better represents the new language use, and is closer to the CLIP image embedding $i$, as illustrated in Figure 2. It is important to note that in order to retain the zero-shot performance, $T$ should in most cases return a similar output as input, unless the text has a domain-specific meaning that the model should correct for.

The transformation function is modelled as $T(t) = t + A(t) \times S(t)$, where $A(t) : \mathbb{R}^{512} \to \mathbb{R}^{512}$ is an **adjustment function**, and $S(t) : \mathbb{R}^{512} \to [0, 1]$ is a **scaling function**. As we will see, this scaling function helps to retain the zero-shot performance of the model. It can be noted that this principle is similar to that of Residual connections in neural networks (He et al., 2016) and Gated Linear Units (Dauphin et al., 2017).

Training examples are stored as pairs of text and image embeddings $\langle t, i \rangle$, and thus have a limited footprint (512+512 floats). $A(t)$ is then trained to estimate the difference vector $i - t$, using the accumulated training examples. We learn $A$ using linear regression: $A(t) = \beta t + m, \beta \in \mathbb{R}^{512 \times 512}, m \in \mathbb{R}^{512}$. To avoid overfitting (given the limited number of training examples) we use ridge regression

---

[1]For the experiments in this paper, we use the publicly released pre-trained CLIP model with the ViT-B/32 Vision Transformer architecture (https://github.com/openai/CLIP).

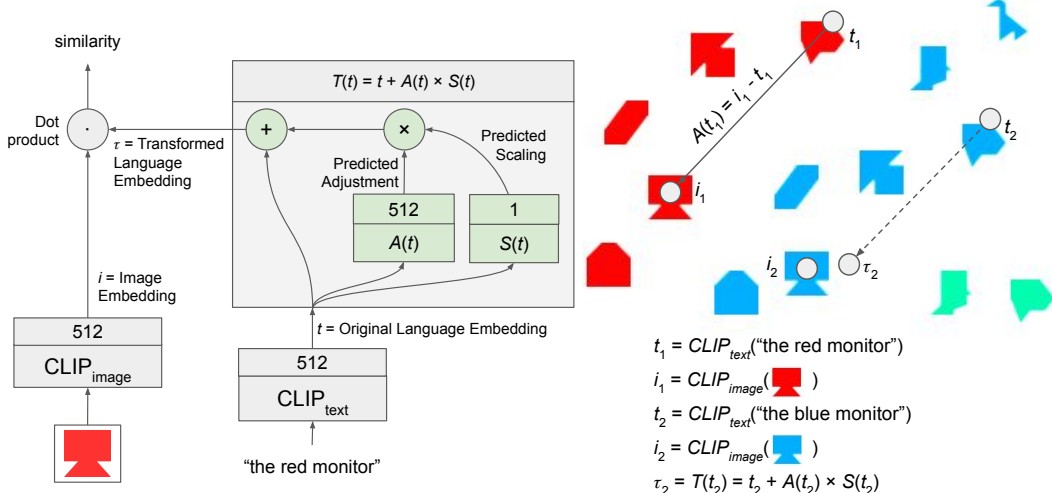

Figure 2: The CoLLIE transformation.

Figure 3: A principled illustrated of the intuition behind CoLLIE.

(L2 regularization with $\lambda = 0.001$). The objective of $S$ is to return a value close to 1 when the input is a text that should be transformed (i.e., close to any example in the training set), and close to 0 otherwise. We learn $S$ using support vector regression (SVR) with an RBF kernel (forced in the range $[0, 1]$). The accumulated training examples are used as positive examples (with training target 1). As negative examples (with training target 0), we simply use a list of the 1000 most common nouns in English (representing expressions that should not be transformed), which seemed to work relatively well in our initial tests.

Figure 3 illustrates the intuition behind the model: Given that we have a reference to an image ("the red monitor"), we encode it using CLIP and get an embedding $t_1$. As can be seen, in this case the text embedding is not very close to the embedding of the corresponding image $i_1$, and will thus retrieve the wrong referent. To teach the model to make better predictions in the future, we add the pair $\langle t_1, i_1 \rangle$ as a training example. Using the accumulated training examples, we train $A$ to approximate the difference vector between the embedding of the image and the text ($i_1 - t_1$). Now, when a new referring expression is to be resolved, "the blue monitor", the expression is encoded by CLIP into $t_2$. Again, directly using this embedding would result in a poor match for this domain. If we now apply the learned adjustment function $A(t_2)$, it is likely to return a similar vector as $i_1 - t_1$ (given that $t_2$ is relatively close to $t_1$ and that we do not have any other, more similar, training examples). Similarly, $S(t_2)$ is likely to return a value close to 1. We now get a new vector $\tau_2 = t_2 + A(t_2) \times S(t_2)$, which is indeed closest to the true referent $i_2$.

While our choice of models for $A(t)$ and $S(t)$ are just two examples of classes of functions that could be used in the CoLLIE transformation, we chose them for demonstrating the efficiency of the approach, even when these functions are very simple. Other adjustment and scaling functions could of course be explored in future work. However, given that we want to be able to learn these functions with very few examples, it is important that we use models requiring relatively few parameters.

## 5 EXPERIMENTS

### 5.1 EXPERIMENT I: LEARNING PSEUDO-WORDS FOR REALISTIC OBJECTS

We first devised an evaluation scheme to see whether the model can learn new words for photographic images from the LAD dataset, and monitor the retained zero-shot performance during training. We randomly select a set of $N$ categories, $C_{train}$, (out of the 200 categories) for which we want to teach the model new names. We then assign a new name for each of these $N$ categories, using randomly selected pseudo-words from the Novel Object and Unusual Name (NOUN) Database ("boskot", "derd", "tust", etc.) (Horst & Hout, 2016). Training is then performed over five

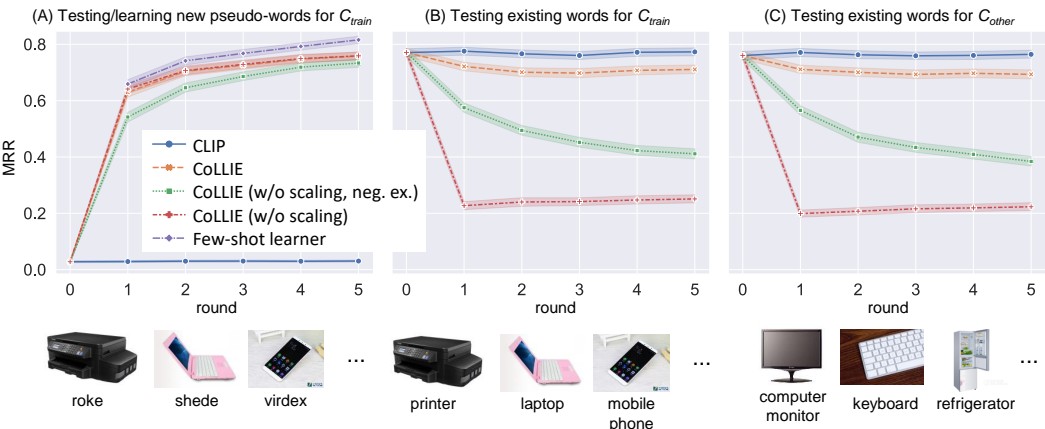

Figure 4: Performance of the model over five rounds of training on the LAD dataset (averaged over 50 iterations, 95% CI), where $|C_{train}| = 50$. One training example per category is added per round. (A) shows continual learning performance. (B) and (C) show retention performance. Few-shot learner is only applicable for pane (A).

and testing is performed over six *rounds* (the initial round of testing is performed without training, which reflects the model's zero-shot performance). At the beginning of each round, we randomly select 1 image for each of the 200 categories, without ever reusing images between rounds. We then let the model rank the 200 images as potential referents for each pseudo-word, and the MRR is computed as described in Section 3. At the end of each round, we add the images from $C_{train}$ and their associated pseudo-words as training examples (i.e., one example per category) to the model, and retrain it. This whole procedure is repeated over 50 *iterations* (with new pseudo-words and categories randomly selected and assigned), in order get a smooth average performance per round. We evaluate and compare the performance using (1) the CoLLIE model, (2) the fixed CLIP model, and (3) a few-shot classifier based on logistic regression (implemented in the same way as in Radford et al. 2021). For the few-shot classifier, each pseudo-word is treated as a class. To study the effect of the scaling function, we also add (4) the CoLLIE model without the scaling function, and (5) the CoLLIE model without scaling function, but where the negative examples (common nouns) are added as training examples to the adjustment function to produce zero-length vectors (i.e., no adjustment).

The results are shown in Figure 4(A), where $N = 50$. As can be seen, CoLLIE quickly learns the new pseudo-words, and reaches a fairly good performance (0.618) already after one round (i.e., when it has only been provided with one example per category), increasing to 0.750 at the final round (where five examples have been provided), which is quite close to CLIP's zero shot performance of 0.773 for the original words on this dataset. Here, the scaling function has very little effect. Since the CLIP model is not doing any learning, it obviously has a very poor zero-shot performance on these new words. However, the few-shot classifier has a slightly better performance than CoLLIE, especially after five examples are added (0.805). This is perhaps not very surprising, given that it is optimizing this classification task, rather than transforming the embedding space. Also, in this specific task, CoLLIE does not benefit from generalization of the learned words, as they are arbitrarily assigned and there is no semantic compositionality effect.

To study the retained zero-shot performance of the model during training, we also plot the performance of the models when using the original words for the 50 categories in $C_{train}$, in Figure 4(B). As can be seen, the original names for those categories can still be resolved by CoLLIE with a slight (but not catastrophic) drop in performance compared to the static CLIP model (0.699 vs. 0.760 at the final round), even though the model has also learned new words for them. In this case, the scaling function is important and without it the performance drops considerably (to 0.260). Similarly, for each round, we also study the models' retained zero-shot performance on 50 randomly selected categories, $C_{other}$, which were not part of $C_{train}$, using their original names. This is shown in Figure 4(C). Again, when using the scaling function, CoLLIE does not seem to interfere much with CLIP's original zero-shot performance (0.719 vs. 0.769 at the final round), while there is a drastic

Table 1: Performance of the model (MRR) on the LAD dataset (averaged over 50 iterations), with different numbers of classes/pseudo-words to learn ($N$). (A) shows continual learning performance and (B) shows retention performance. n/s = negative samples used for the scaling function.

| $N = |C_{train}|$ | 10 | 50 | 100 | 150 | 10 | 50 | 100 | 150 |
|---|---|---|---|---|---|---|---|---|
| **(A) New pseudo-words** | $C_{train}$: After 1 example | | | | $C_{train}$: After 5 examples | | | |
| CoLLIE (1000 n/s) | 0.616 | 0.618 | 0.636 | 0.634 | 0.770 | 0.750 | 0.746 | 0.751 |
| CoLLIE (100 n/s) | 0.639 | 0.629 | 0.639 | 0.636 | 0.770 | 0.750 | 0.746 | 0.752 |
| CoLLIE (w/o scaling) | 0.654 | 0.632 | 0.641 | 0.637 | 0.774 | 0.751 | 0.746 | 0.753 |
| Few-shot learner | 0.639 | 0.648 | 0.660 | 0.660 | 0.802 | 0.805 | 0.810 | 0.815 |
| CLIP | 0.033 | 0.030 | 0.030 | 0.028 | 0.028 | 0.029 | 0.030 | 0.031 |
| **(B) Original words** | $C_{train}$: After 5 examples | | | | $C_{other}$: After 5 examples | | | |
| CoLLIE (1000 n/s) | 0.790 | 0.699 | 0.606 | 0.519 | 0.761 | 0.719 | 0.608 | 0.515 |
| CoLLIE (100 n/s) | 0.769 | 0.621 | 0.450 | 0.333 | 0.732 | 0.638 | 0.443 | 0.314 |
| CoLLIE (w/o scaling) | 0.515 | 0.260 | 0.127 | 0.078 | 0.361 | 0.242 | 0.112 | 0.054 |
| CLIP | 0.798 | 0.760 | 0.768 | 0.773 | 0.779 | 0.769 | 0.783 | 0.774 |

drop in performance (to 0.242) when the scaling function is not used. Since the few-shot learner has no zero-shot performance to retain, its performance is not plotted in pane B-C.

As these experiments show, the scaling function is important for the performance of the model, and simply adding the negative examples to the adjustment function does not have the same effect (as seen in Figure 4). In the Appendix (Table 2), we also investigate different implementations of the scaling function, of which SVR has an overall favorable performance.

To further investigate the performance, we also run experiments with different numbers of classes/pseudo-words to learn ($N = |C_{train}|$), and different numbers of negative examples in the scaling function. The results are shown in Table 1. As can be seen in (A), the continual learning performance is relatively stable for different values of $N$. However, as seen in (B), the zero-shot retention performance is clearly affected as $N$ increases. This is especially true when only 100 negative examples are used for the scaling function. Thus, it is likely that the drop in retention performance could be mitigated by adding even more negative examples, as $N$ increases. As seen in (A), the number of negative examples does not seem to have a big effect on the continual learning performance.

## 5.2 Experiment II: Learning language for tangram figures

The biggest expected benefit of CoLLIE comes from its ability to generalize from the language it is learning, which was not addressed in Experiment I. We thus devised an evaluation scheme to see how quickly the model can learn to identify the colored tangram shapes (introduced in Section 3), for which it clearly had a very poor zero-shot performance. Here, we expect the model to benefit from the compositionality of the referring expressions. As discussed earlier, given that it has learned what concept to associate with the phrase "the blue rock", it should be able to extrapolate this understanding to "the red rock".

Again, the task is to rank the 85 potential referents, given a referring expression. Similar to Experiment I, the model starts out with no training examples (round 0). We then train the model over 30 *rounds*. In each round, one random referent is picked, the model's performance (in terms of MRR) on this referent is assessed, the image-text pair of the referent is added to the training set, the model is retrained, and a new round begins. This whole procedure is then repeated over 3000 *iterations*, resetting the model after each iteration. The MRR per round (over all iterations) is illustrated in Figure 5. We do a similar comparison with other models as in Experiment I. Here, we let the few-shot learner (again, a logistic regression classifier) fall back on CLIP when it is presented with a referring expression it has not seen before. We then add the new referring expression as a new class for the few-shot learner, and retrain it.

As can be seen, CoLLIE quickly learns the names for the tangrams, reaching an MRR of 0.860 after 30 rounds. Note that the 85 images are all unique in terms of shape-color combinations, which means that the model must be able to generalize in order to achieve this performance. In

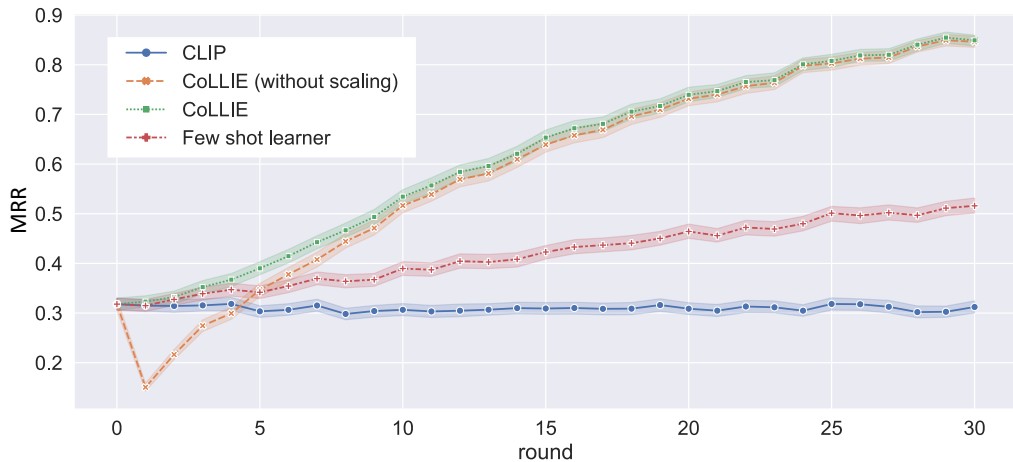

Figure 5: Performance of the model over 30 rounds of training (averaged over 3000 iterations) when training on the colored tangrams (95% CI). One new training example is added per round.

contrast, the few-shot learner has a much worse performance, as new referring expressions (classes) are introduced in most rounds in the beginning (unless the exact same object happened to be picked twice), and it has no way of generalizing from already learned classes. This confirms the hypothesis that CoLLIE should be able to benefit from the compositionality of language: After being taught what a "red giraffe" looks like, CoLLIE is now better at identifying a "blue giraffe", combining the base representation of "blue" with the learned meaning of "giraffe".

To further confirm this ability, we also performed an experiment where we first train the model on all 17 shapes of one random color, and then evaluate it on the same shapes with different random colors. This was iterated 100 times. Whereas the CLIP baseline model (and the few-shot learner, which has to fall back on the CLIP model) only had an average MRR of 0.317 on these unseen combinations, CoLLIE achieved an MRR of 0.857.

The intuition behind why this works was illustrated in Figure 3: CoLLIE learns to predict the *difference vector* that needs to be applied. Thus, if the color dimensions in the CLIP embedding were already aligned between the language and the image, there will not be any need to adjust those dimensions – it is only the dimensions related to the shape that need to be adjusted. The fact that this works despite CLIP's representation being entirely distributed is interesting. The steady improvement also indicates that the learning of each concept does not interfere with learning of other concepts. However, as can be seen, without the scaling function, the model has a drastic drop in performance for the first rounds, which is likely because the newly learned adjustments are added too generously to unrelated referring expressions.

As a further (limited) test to verify the model's ability to generalize, we substituted the names of the tangrams with synonyms[2] ("barn"→"shed", "chicken"→"hen", etc.). This way, we formed referring expressions such as "the blue hen". Using the CoLLIE model trained for 30 rounds as described above, we then evaluated these expressions (over all 3000 iterations). The MRR for these was 0.602, which is clearly better than the baseline of 0.290 (using CLIP directly), providing further evidence for the model's ability to generalize. Given that many of the names had no obvious synonyms, the individual performance of them varied greatly (MRR 0.056-0.920). A breakdown of these results can be found in the Appendix (Figure 7).

## 6 DISCUSSION

Several previous studies have addressed the problem of incremental class learning in image classification (Kemker & Kanan, 2018; Kemker et al., 2018). However, to the best of our knowledge, the

---

[2]Taken from `thesaurus.com`

problem of continual adjustment of language-image embeddings to learn new language grounding has not been addressed before. Thus, we do not have any results from prior work to compare our performance with. However, we hope that this work can serve as a benchmark for future studies and alternative models.

Returning to the four characteristics we aimed to achieve, we think that the model has shown to be **sample efficient**, as it seems to reach a fairly high performance with only one training example per new category. Second, Experiment II showed that the model was able to **generalize** from the newly learned language use, thanks to the semantic compositionality of the referring expressions. Third, Experiment I showed that the model was fairly **robust**, as despite a slight drop in the model's original zero-shot performance, it did not exhibit catastrophic forgetting. Finally, the model is fairly **computationally efficient**, seeing as the transformation model uses very simple models with few parameters and the stored training examples have a very small footprint. Nevertheless, the transformation model needs to be retrained when new examples are added, so there are limits to its scalability. Whether this is a problem, however, depends entirely on the use case scenario. Regardless, the continual learning of the transformation function without storing examples is also an interesting topic for future work.

As we have seen, the scaling function plays a very important role in retaining the model's zero-shot performance, making sure that only the newly learned terms are adjusted. However, given how the scaling function was trained here (simply using common nouns as negative examples), this will not always work, and we therefore still saw a slight drop in zero-shot performance, especially as the number of new concepts to be learned increases. The scaling function could of course be more or less restrictive. For example, it could require an exact match with a training example to set the scale to 1, and 0 otherwise. This would retain all of the zero-shot performance, at the expense of being able to generalize the learning to similar language use. Exploring more sophisticated scaling functions that provide a good balance between retention and generalization is an interesting topic for future work. For example, Shin et al. (2017) explore the use of a generative model to generate samples for rehearsal, which alleviates the need for storing training examples.

Of course, CoLLIE's performance also relies on CLIP already having good representations of the "new" categories, and so it can be argued whether the categories themselves are really new – it is rather the transformed embedding of the language that better maps to this region in the embedding space. In other words, it might be fair to say that the model learns domain-specific *language use*. CoLLIE's performance is thus limited by the performance of the base model (CLIP in this case) to accurately represent the landscape of visual properties of objects. As pointed out by Radford et al. (2021), CLIP's representations are limited in certain aspects, including counting objects in an image or representing detailed attributes.

An interesting topic for future work is how to consolidate the learned transformation function into the base model, and then learn a new transformation function on top of this, or to use different transformation functions in different contexts, as language use is highly context dependent. Another line of future work is to incorporate the model into a system that learns through interaction. Given the small number of examples needed to learn new language use, the model should be interesting for studies on continual language grounding in the context of human-robot interaction (e.g., Chai et al. 2016).

## 7 CONCLUSION

We have presented CoLLIE: a simple, yet effective model for continual learning of how language is grounded in vision. Given a pre-trained language-image embedding model capable of zero-shot image classification (CLIP), CoLLIE learns a transformation function that adjusts the language embeddings when needed to accommodate new language use. The transformation function learns the difference vector that needs to be applied to the embedding, and uses a scaling function to retain embeddings that should not be adjusted. We establish new benchmarks with novel metrics to capture the trade-off between continual learning, retention (avoiding catastrophic forgetting), and generalization. The evaluation showed that the model can learn new language use with very few examples. Unlike traditional few-shot learning, the model does not just learn new labels, but can also generalize to similar language use, and benefit from the semantic compositionality of language.

## 8 REPRODUCABILITY STATEMENT

The models were implemented using scikit-learn (`https://scikit-learn.org/`), with standard parameters unless otherwise stated. The code used for running the experiments and reproducing the results in this paper is provided as supplementary material, including necessary data or pointers to data.

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

## A  APPENDIX

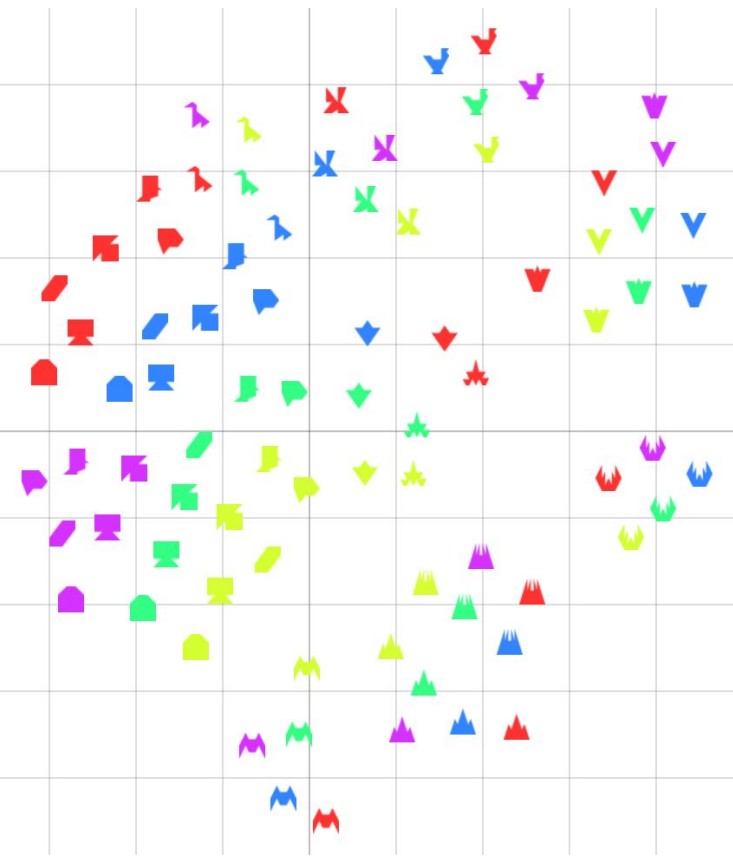

Figure 6: t-SNE dimensionality reduction of the colored tangram CLIP embeddings.

Table 2: Performance of the model (MRR) on the LAD dataset (averaged over 50 iterations), with different implementations of the scaling function, where $|C_{train}| = 50$.

|  | Learning new pseudo-words for $C_{train}$ | | Testing existing words for $C_{train}$ | | Testing existing words for $C_{other}$ | |
|---|---|---|---|---|---|---|
| Round | 1 | 5 | 1 | 5 | 1 | 5 |
| SVR (RBF) | 0.630 | 0.758 | 0.721 | 0.711 | 0.711 | 0.693 |
| SVR (sigmoid) | 0.574 | 0.756 | **0.759** | 0.690 | **0.749** | 0.672 |
| SVR (linear) | 0.614 | 0.758 | 0.736 | 0.684 | 0.729 | 0.667 |
| SVR (poly) | 0.635 | 0.758 | 0.711 | **0.715** | 0.701 | **0.698** |
| Logistic regression | 0.633 | 0.756 | 0.656 | 0.679 | 0.649 | 0.657 |
| Linear regression | **0.640** | **0.759** | 0.646 | 0.644 | 0.640 | 0.624 |

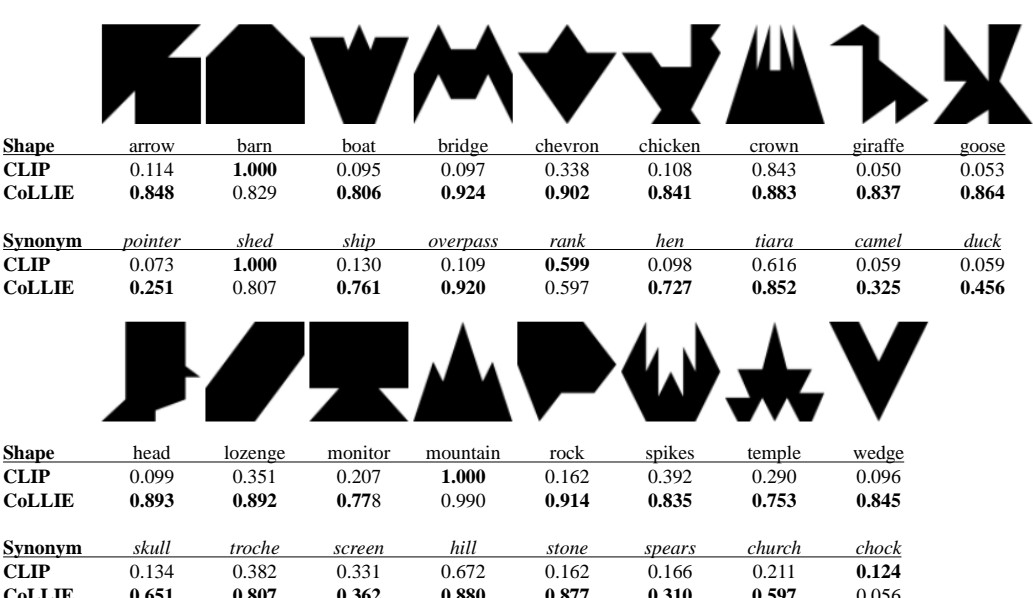

| Shape | arrow | barn | boat | bridge | chevron | chicken | crown | giraffe | goose |
|---|---|---|---|---|---|---|---|---|---|
| **CLIP** | 0.114 | **1.000** | 0.095 | 0.097 | 0.338 | 0.108 | 0.843 | 0.050 | 0.053 |
| **CoLLIE** | **0.848** | 0.829 | **0.806** | **0.924** | **0.902** | **0.841** | **0.883** | **0.837** | **0.864** |

| Synonym | *pointer* | *shed* | *ship* | *overpass* | *rank* | *hen* | *tiara* | *camel* | *duck* |
|---|---|---|---|---|---|---|---|---|---|
| **CLIP** | 0.073 | **1.000** | 0.130 | 0.109 | **0.599** | 0.098 | 0.616 | 0.059 | 0.059 |
| **CoLLIE** | **0.251** | 0.807 | **0.761** | **0.920** | 0.597 | **0.727** | **0.852** | **0.325** | **0.456** |

| Shape | head | lozenge | monitor | mountain | rock | spikes | temple | wedge |
|---|---|---|---|---|---|---|---|---|
| **CLIP** | 0.099 | 0.351 | 0.207 | **1.000** | 0.162 | 0.392 | 0.290 | 0.096 |
| **CoLLIE** | **0.893** | **0.892** | **0.778** | 0.990 | **0.914** | **0.835** | **0.753** | **0.845** |

| Synonym | *skull* | *troche* | *screen* | *hill* | *stone* | *spears* | *church* | *chock* |
|---|---|---|---|---|---|---|---|---|
| **CLIP** | 0.134 | 0.382 | 0.331 | 0.672 | 0.162 | 0.166 | 0.211 | **0.124** |
| **CoLLIE** | **0.651** | **0.807** | **0.362** | **0.880** | **0.877** | **0.310** | **0.597** | 0.056 |

Figure 7: Performance (MRR) on individual tangram shapes and their synonyms. Both the original zero-shot performance of CLIP (over the 85 candidates), and the performance of CoLLIE (after training on 30 examples with the original names), are shown.

