# OpenReview forum: "CoLLIE: Continual Learning of Language Grounding from Language-Image Embeddings"
_ICLR.cc/2022/Conference — ICLR 2022 Submitted_

### Official Review · Reviewer_tBdi · 2021-10-31

**Correctness:** 3
**Technical Novelty And Significance:** 2
**Empirical Novelty And Significance:** 2
**Recommendation:** 3
**Confidence:** 3

**Main Review:**

Strength: the experiment shows that a small network and a simple difference vector are enough to achieve similar performance to the designed few-shot learner.

Weakness: This paper tries to use a small network to achieve continued learning. The main idea to achieve it is to predict a difference vector between exit-word and pseudo-word. This idea is somewhat similar to the idea of GLoVe, which also has a similar difference vector between multiple word pairs. From the experiment, the idea works for the 2 test cases. However, the test data is constructed (like “the red monitor" and “the blue monitor") and thus the reference image-language pairs can easily be found. I wonder that if the text queries are more complex, like a long sentence or a paragraph, will the idea still work. Usually, in a real case, a more complex text is more common than well-defined constructed phrases or words.

In the experiment, the author mentions a few-shot learner, which is not fully described in the paper. How does the author designs the few-shot learner may also influence the performance? Few-shot learning is now a popular topic and many papers are proposed. I think more experiments for the different few-shot methods on the test dataset will be more convenient.



**Summary Of The Paper:**

This paper target on continual learning for the language-vision ground task with an additional small network. The network is attached on a CLIP model and it contains 2 linear layers to predict a difference vector of the original text embeddings and scaling value. The network are optimized by support vector regression (SVR). The method is test on pseudo-word and  tangram figures tasks. The result shows the model with proposed method can achieve better performance of original CLIP model.


**Summary Of The Review:**

I think the idea of the difference vector is similar to some previous work. For the experiment, the data is somewhat constructed and simple for the method to find the relative object or description. However, It does not show that if the method can also be applied to the more complex case, which may be more common in the real life. And the comparison is not so sufficient. The author may compare with more few-shot methods to support their contributions.

In general, I think the paper still needs to be improved and I recommend rejecting it this time.

---

> ### Author Response · Authors · 2021-11-10
> **Replies to your concerns**
>
> Thank you for your helpful review. Regarding your concerns:
>
> * We are not sure we understand your comparison with GLoVe. We have not seen experiments with GLoVe where they try to learn difference vectors with GLoVe (other than illustrating things like king-man+woman=queen)?
>
> * It is true that our texts are quite short, but the aim here is to be able to model referring expressions, and they are typically not in the form of long sentences or paragraphs, but rather noun phrases. As we operate on the CLIP embeddings, the issue is of course whether CLIP can represent longer texts well or not (i.e., represent the different semantic nuances in the text). In our experience, this does not seem to be the case (which is likely explained by limitations in CLIP’s training data). If so, CoLLIE would then of course not be able to make use of longer texts either. If this is seen as a limitation of our model, then it would be fair to also hold that against CLIP as a model, as it was also evaluated with fairly short texts. As acknowledged in the CLIP paper (and in our discussion), it does have problems with for example negations and spatial relations.
>
> * As described in the paper, the few-shot learner is a multi-class logistic regression classifier. We used the exact same few-shot classification setup as the one used in the original CLIP paper.

---

### Official Review · Reviewer_qatX · 2021-11-03

**Correctness:** 2
**Technical Novelty And Significance:** 2
**Empirical Novelty And Significance:** 3
**Recommendation:** 5
**Confidence:** 5

**Main Review:**

This paper is unique in its problem setting because it combines continuous learning with language and image embedded representations. Although the proposed method is simple, the experimental results show its ability to deal with new categories while maintaining the original performance.

On the other hand, this paper adopts only CLIP as an existing representation learning method. Although CLIP is a modern and excellent representation learning model, the authors should have conducted similar experiments with other representation learning to show that the superiority of the proposed method is a general contribution to existing representation learning methods. Otherwise, the advantage of the proposed method may exist only when applied to CLIP.

In addition, the proposed method is in the form of a product of an adjustment function that performs a linear transformation and a scaling function that outputs [0,1], which is added to the original representation. It is just a special form of the combination of Gated Linear Units (GLU) (Dauphin et al., 2017) and residual connections. The authors state that the proposed method is simple, but they do not mention that they propose the same as GLU.

Furthermore, the adjustment function and the scaling function are trained with different regression methods. However, there is no explanation why the scaling function, in particular, was implemented with support vector regression. In linear support vector regression, the output of [0,1] as defined by the authors as S(t) cannot be satisfied, and negative values and values larger than one may be output. As described above, the Transformation module is regarded as a combination of the GLU and the residual connection. Similar effects may have been obtained when the module is trained so that the vector representation of the text in the negative sample is output as it is and that the vector representation of the text in the positive sample matches the vector representation of the image. It is unclear why ad hoc learning as proposed is necessary.

Finally, as the authors also state, the rationale for learning the 1000 most common nouns as negative examples is questionable. It may have worked in this case, but what would happen if these nouns were also included in the Few-Shot learning text?

The following are minor concerns.
- The proposed method's behavior would have been better understood if the output of the scaling function alone had been reported, along with Figure 5, for the new pseudo-words and the existing words.
- The authors claim that the experimental results in section 5.2 show the hypothesis that the proposed method can learn blue giraffes better if it learns red giraffes. However, section 5.2 only shows that MRR as a macroscopic retrieval accuracy has improved. Other experiments and evaluations are needed to show whether this hypothesis truly holds. For example, we could train the few-shot training with only red giraffes and then evaluate whether the blue giraffes can be retrieved in zero-shot.
- Finally, CoLLIE is only described in this abbreviation, but its official name is unknown.

**Summary Of The Paper:**

This paper proposes a simple new method for continuous learning of language-image embeddings. The proposed method, CoLLIE, employs a Transformation module designed to work only on the samples undergoing few-shot learning. It consists of a scaling function that determines whether the text is close to the sample undergoing few-shot training and an adjustment function that estimates the difference between the embedded representation of the paired text and the embedded representation of the image. The experimental results show that the proposed method can deal with new categories by few-shot learning while retaining the zero-shot learning performance of existing embedded representations (CLIP).

**Summary Of The Review:**

### Original Summary
This paper proposes a simple and effective method for a unique problem setting. On the other hand, the novelty of the method is much less than the authors may think, and a similar method is not cited. The rationality of the design of the proposed method is also questionable, and its versatility has not been experimentally demonstrated. The reviewer recommends that the authors expand the experiments and resubmit the revised paper to another international conference.

### Final Rating
The reviewer has read the other reviews, the responses from the authors, and the revised manuscript. The first score (3) has been upgraded a little to (5), but the reviewer still leans to reject the paper. The reasons are described in the last post of this thread.

---

> ### Author Response · Authors · 2021-11-10
> **Replies to your concerns**
>
> Thank you for your helpful review. We are glad you found the paper unique in its problem setting. Regarding your major concerns:
>
> * Comparison with other representation learning methods: We do not think there is space in the paper to introduce comparisons with other representation learning methods than CLIP (even if that would certainly be interesting), and we prioritized other experiments. We do acknowledge in the paper though that the performance of CoLLIE relies on the performance of the underlying embedding model (CLIP in our case), since it needs to have good representations of the images, even if the image-language mappings are not appropriate and need to be adjusted through continual learning.  However, when this is the case (as in our experiments), we think that CoLLIE shows promising performance. If other models would be used, with worse performance than CLIP, our performance would of course also go down.
>
> * Regarding comparison to other methods: It is true that the transformations in CoLLIE are similar to those used in GLU and residual connections, and we will add this acknowledgement to the paper. However, we do not think that this affects the novelty of the proposed approach, as we use these mechanisms for a quite different problem.
>
> * Use of SVR: We did not have space in the paper to systematically compare and report different models for the adjustment and scaling functions. However, in our experiments, SVR performed better than for example logistic regression. When looking at the output of the scaling function, we have not seen values below 0 or above 1, but it would of course be possible to bound them in the range of [0,1] to avoid this problem.
>
> * Need for a separate scaling function: In fact, we first tried to use the negative examples directly for the adjustment function (with the target of outputting a zero-length vector), but this did not yield a good performance. We do not have a good explanation at this point why this was the case.
>
> * Use of common nouns as negative examples: If one of these nouns would appear in the new words to be learned, the scaling function would likely output something like 0.5. That would still give some transformation, but it is of course not ideal. We agree that this is not an optimal solution, and we think finding other solutions is an interesting topic for future work.
>
> * Regarding compositionality: It is impossible to get the learning curve of CoLLIE, as seen in Figure 6, if it was not able to make use of compositional semantics (i.e., generalize blue giraffes from red giraffes), since there are 85 unique color/shape combinations. The comparison with the few-shot learner in the same figure (which does not have this compositionality) makes this clear.
>
> * What the CoLLIE acronym stands for can be found in the title of the paper, but we can add this in the text as well.

---

> > ### Comment · Reviewer_qatX · 2021-11-16
> > **The reviewers would like to acknowledge the authors for their responses**
> >
> > - The reviewers believe that there is plenty of space for representation learning methods other than CLIP. For example, it is obvious what Figure 2 and the paragraph about it are trying to convey. The width of Figure 6 is redundant for the message the figure is trying to convey. The authors could have simply moved these to the supplementary material in some cases, or they could have included the results of experiments with other representation learning methods other than CLIP in the supplementary material as well. The answer that it is a matter of space is not convincing.
> > - The reviewer appreciates the fact that the authors admit that the proposed method is similar to GLU. The authors claim that the novelty is not impaired because they use it for a different problem, but this is an overstatement. There is a novelty in applying an existing method to a different problem, but it is impossible to claim that it is the same novelty as proposing a method that does not exist at all.
> > - It is unfortunate that the defense of the use of SVR without explanation of the rationale is also simply a matter of space. It would have been much better if the author had simply stated what other methods were compared, and SVR was the best by showing a performance comparison.
> > - For the separate scaling function, if the proposed method was the best, the authors should have shown what else they tried. Again, "not enough space" is not a reason to give up adding that level of description.
> > - The author would like to appreciate that the authors agree that the use of common nouns as negative examples is a necessary improvement. ICLR is a top-level international conference, but some limitations should be welcomed as rather important information if there are essential contributions.
> > - Regarding compositional semantics. The authors continue to claim that the proposed method is able to learn compositional semantics only from macroscopic evaluation, but the reviewer disagrees. It is possible that the baseline does not work well due to the domain gap between CLIP and tangrams, or the proposed method may have better accuracy for other reasons. In other words, the authors' explanation is not a necessary and sufficient condition. If they wanted to claim that "the proposed method can learn blue giraffes better if it learns red giraffes", it would have been faster to show examples of actual behavior using those tangrams.
> > - The reviewer feels it odd that the title "CoLLIE: Continual Learning of Language Grounding from Language-Image Embeddings" gives the full name of CoLLIE; it would have made sense if the name of the method were CoLLGLIE.

---

> > > ### Author Response · Authors · 2021-11-22
> > > **Response to reviewer**
> > >
> > > Thank you for your answers. We think that we have now addressed all of your concerns, except the first one, in the updated version of the paper (see the general comments). Regarding your first concern, we have not found any other representation learning models than CLIP, which are both generally available and have comparable performance with CLIP. If you have any specific models in mind, we would be very happy to know about them. Remember that we cannot use any representation learning method for images, but something much more specific, that can learn to embed texts and images in the same space. One example would be ALIGN by Google AI, but it does not seem to be generally available.

---

> > > > ### Comment · Reviewer_qatX · 2021-11-30
> > > > **Response to author**
> > > >
> > > > The reviewer would like to thank the authors for their response and revised manuscript. After careful consideration, the reviewers has raised the rating slightly.
> > > >
> > > > This is because:
> > > > - the authors have referred to GLU.
> > > > - the authors have added some experimental results.
> > > >
> > > > However, the final score is still leaning to reject; the reasons are as following:
> > > > - As the authors stated, the proposed method is very close to GLU. The novelty is the way to use such module in a deep neural network. The technical contribution are not significant.
> > > > - The defense of the fact that the authors only report the results of the CLIP experiment is inconsistent and inadequate.
> > > >   - First, the authors initially stated that they were only able to include the results of the CLIP experiment on the basis of paper space limitations. As the reviewer pointed out, the authors could have reported the results of other experiments. In fact, the authors have added some experimental results in the revised manuscript. The authors then insisted the lack of a suitable method other than CLIP as the reason. The space limitations and the lack of appropriate methods are two completely different perspectives, and the authors seem to be blurring their argument.
> > > >   - Moreover, the reviewers do not think that there is no appropriate method other than CLIP, as the authors say. In the first place, it is not a reason that ALIGN cannot be experimented with because it does not release its source code. Besides, learning to embed images and text in a common space has been widely studied before in the field of cross-modal retrieval. There are so many methods so that the reviewer could not mention all of them, but some of the following also provide their source codes.
> > > >
> > > > 1. Frome et al., DeViSE: A Deep Visual-Semantic Embedding Model. NIPS, 2013.
> > > > 1. Kiros et al., Multimodal Neural Language Models. ICML, 2014.
> > > > 1. Faghri et al., VSE++: Improving Visual-Semantic Embeddings with Hard Negatives. BMVC, 2018.

---

> > > > > ### Author Response · Authors · 2021-12-06
> > > > > **Response to reviewer**
> > > > >
> > > > > We would like to thank the reviewer again for the answer and for raising the score. We would still like to comment on the two concerns that are left.
> > > > >
> > > > > * We do acknowledge that the method bears some similarities to GLU and residual connections in its use of a scaling function (similar to a gating function). However, we think the mechanism is applied very differently here, on top of the model’s embeddings and trained separately, rather than trained together with the rest of the deep neural network. We don’t think that once a general concept such as gating functions has been used once, no more papers using a similar mechanism (in whatever way) can be considered novel, especially when considering both technical and empirical novelty.
> > > > > * Regarding comparison with other base models than CLIP, we don’t think we are “blurring the arguments”. We believe two things can be true at the same time: (1) We do have space limitations, and (2) it is hard to find **generally available** models that have **comparable performance with CLIP** (as we wrote). We are not sure whether the reviewer actually suggests that we are expected to train the Google ALIGN model ourselves from scratch, but we don’t think that would be feasible. The available models suggested by the reviewer are all several years old, and while we could have applied CoLLIE to those, we do not think this would have added much value to the paper. Since these models no longer have anywhere near SotA performance on relevant downstream tasks, we expect few meaningful insights to be derived from comparing their performance on, for example zero-shot retrieval on the LAD dataset and the subsequent effects of applying CoLLIE, with that of models that exceed their performance by a large margin, such as CLIP (note that the authors of CLIP also did not make any such comparison with these earlier embedding models in their evaluation). Also, they would most likely not be able to make useful representations of the pseudowords we use (as they require sub-word representations). If CoLLIE would not be able to adjust those language/image representations, it is not clear what conclusions we could draw, since CoLLIE requires the representations themselves to be good (even if the image/language mappings need to be adjusted), as we point out in the paper. Thus, if we would include those comparisons in the paper, we don’t think it would have been enough to just add a table with numbers, we would also have to go in depth and analyse **why** they don’t work. And we don’t think we have space for that in the paper (but it could be an interesting analysis for future work). We have already moved several results to the appendix (as suggested by the reviewer), and we think there is a limit to how much central material can be put there.

---

### Official Review · Reviewer_jzf6 · 2021-11-03

**Correctness:** 3
**Technical Novelty And Significance:** 2
**Empirical Novelty And Significance:** 2
**Recommendation:** 5
**Confidence:** 4

**Main Review:**

# strengths and weaknesses
- Strengths
   - Clear and thoughtful discussion of the motivation, namely the importance of grounded language learning and continual learning. I enjoyed reading it!
   - Straightforward description of the main technical contribution
 - Weaknesses
   - As presented, I'm not sure if the presented approach has signicant technical novelty. The proposed solution boils down to finding a linear transformation for the frozen CLIP representations. I would argue that this is already present in any system that uses a foundational model, e.g., BERT or CLIP, to produce input representations for a downstream task (see [1] for broad overview of typical practices). It's common practice for the model to include a fully-connected feed-forward network for transformation of these input embedding.
   - As presented, I don't think the given approach can properly be called "continual learning". CoLLIE re-purposes the CLIP representations for a specific task, but the CLIP model itself is unchanged, and no transfer-learning or fine-tuning happens on the CLIP weights. In sec. 6 its mentioned that future work could explore incorporating the "learned transformation function into the base model." I agree that this direction is worth exploring and I think it would yield results that would better fit the description of "continual learning".
   - In a similar vein, it's unclear how this approach could be scaled to multiple tasks, which is a vital part of any continual learning system (see comments below)

# Comments and clarifying questions
- sec. 4: Regarding the function $A(t) = \beta t + m$, could the authors give a brief explain for their choice of a affine transformation function here? Was this motivated by some previously observed property of the CLIP embedding space? Why should an affine transform be preferred over any other non-linear transform, e.g. quadratic, affine with a non-linear activation etc.?
- sec. 4: Can the authors comment on how this approach can be scaled to deal with multiple distributional shifts? What happens if the descriptions for some of the tangrams change? Would you propose creating a second $A'$ and $S'$? If so, how would they interact with $A$ and $S$?
- sec 5.1: What is the "few shot classifier"? If it is a multi-class classifier, then how can it be compared to CoLLIE, which is learning a fundamentally different task, namely a mapping between embedding spaces?
- sec 5.1: Shouldn't the CoLLIE (without scaling) be trained with negative examples in order to give a fair comparison? More exactly, the paper mentions an ablation where the classifier $S(.)$ is removed from the system, so the system only consists of the transformation $A(.)$. Figure 5 is claimed to show that $S(.)$ is necessary to prevent catastrophic forgetting. However, the training for the ablated system only consists of (pseudo-words, image) pairs. So the transformation $A$ does not have any negative examples, which $S$ would normally have access to.
- sec 6: "stored training examples have a very small footprint": If I understand correctly, these training examples are not stored in memory during prediction, so is it very important that their footprint be small?

# Suggestions
- Sec 1: "Unless the model is re-trained entirely from scratch (which is infeasible for large models like CLIP), the updates to the model’s parameters might interfere with previously learned knowledge, resulting in abrupt performance drops."
  - As I understand, this reasoning explains why transfer-learning or fine-tuning on CLIP is undesirable. However, is there reason to believe that catastrophic forgetting will occur for the settings considered in this paper (nonce-words, tan-gram descriptions)? The tasks don't seem too dissimilar from what CLIP sees ordinarily during training. I think it would strengthen your case if Figure 5 or Table 1 had results that show how forgetting occurs when the original embedding model is fine-tuned. I understand that CLIP isn't available for fine-tuning right now, but maybe a conceptually similar multi-modal transformer could be used.
- Sec 3: Could you report the mean squared error between the transformed vectors $T(t)$ and the target vectors?
  - I ask because it's possible that the similarity ranking for a given embedding may be high relative to the other candidates, but could still be a large distance away from the target.
  - In that vein, do you have a sense of what images your transformed embeddings correspond to? Can images be decoded for them?
- Sec 5.2: It is claimed that the results in figure 6 demonstrate that the model can leverage compositional knowledge to generalize. Given that results are averaged over 3000 runs, I am inclined to believe this claim. But it could also be shown explicitly by creating a held-out test-set of unseen color+shape combinations. This would increase confidence in the claim that the model is exhibits some type of systematic generalization.

# citations
[1] Bommasani, Rishi, et al. "On the opportunities and risks of foundation models." arXiv preprint arXiv:2108.07258 (2021).
[2] Magassouba, Aly, Komei Sugiura, and Hisashi Kawai. "CrossMap Transformer: A Crossmodal Masked Path Transformer Using Double Back-Translation for Vision-and-Language Navigation." arXiv preprint arXiv:2103.00852 (2021).


**Summary Of The Paper:**

This paper presents a way to take general-purpose joint text+image representations, e.g., from CLIP, and transform them for use on a new task, while still preserving performance on the old task.

The gist of the idea is this: given a new task, a classifier is trained to decide whether or not a given input belongs to the new task. If it does, then that input's representation is transformed according to an affine transformation.

More exactly, the proposed approach can be broken into two pieces. The first piece is an affine transformation $A$ that is used to transform an original CLIP embedding $t$ into a new representation $A(t)$ which is better suited for the new task. The second piece is a scaling function $S(t):\mathbb{R}^{512}\rightarrow [0,1]$ that predicts how likely it is that the input $t$ belongs to the new task. For a given input $t$, the transformed vector is then given by $t+A(t)\times S(t)$.

Two experiments are described: (1) learning to associate nonce word descriptions with photographic images and (2) learning to associate descriptions with colored tan-grams which were not seen by CLIP in pre-training.

For both experiments, $S$ is a support vector machine that is trained to identify whether a given descriptions belongs to the new task, and $A$ is trained using linear regression. Performance is scored based on how highly the target image is ranked among a set of other images, where the ranking is done by dot-product similarity.

In both experiments, they find that their system performs well on the new task, while still retaining the performance on other examples.

**Summary Of The Review:**

Overall, the paper is well-written and outlines good directions for future work. The motivations are described clearly and thoughtfully. However, as is, I do not think it represents a significant enough contribution. The approach described is already present in most systems that use a large pre-trained model to provide embeddings for the input. That is, it's common for most existing approaches to take an off-the-shelf model, freeze the weights, and then use the produced embeddings for some down-stream task (see for example [2]). Finally, I'm not sure if the proposed solution can properly be called "continual learning" since it's unclear how it should scale with the addition of new tasks.

---

> ### Author Response · Authors · 2021-11-10
> **Replies to your concerns**
>
> Thank you for your thorough and insightful review. We are glad you enjoyed reading the motivation and found the description of the approach easy to follow. Regarding your concerns:
>
> * Technical novelty: It is true that the approach boils down to finding a linear transformation for the frozen CLIP representations, and described in that way CoLLIE is certainly not novel. In fact, we try our best to explain that standard few-shot learning (e.g. logistic regression) is also trained on top of frozen representations. However, we are not aware of any previous work that learns a direct transformation of the representations themselves to accomplish both continual learning and retained zero-shot use of those representations with the same model. We think it is fair to characterize the output of the model as exhibiting continual learning, even if the CLIP representations are frozen. The motivation for freezing the CLIP representation is of course that it would be very hard to achieve the objectives set out in the introduction of the paper, if the entire model would be retrained. As we describe in the paper, there are other continual learning models that have been proposed where learning only happens in parts of the model.
>
> * We are not sure we understand what you mean by scaling the approach to “multiple tasks”? In our paper, the “task” is to adjust the language/image mappings in a multimodal representation model to accommodate new language use. Finding out whether the model could be used for other tasks is beyond the scope of this paper.
>
> * Choice of affine transform: This is a simple transformation with relatively few parameters, that can be trained with very little data, and corresponds to a single layer in a neural network. We did not have space in this paper to systematically experiment with different transformations, but in our tests, this was the best transformation we could find.
>
> * Multiple distributional shifts: In general, the representations that are shifted are scattered across the embedding space, and there is no systematic way in which these transformations happen between different language-image pairs. Thus, we have no reason to believe that the model should not be able to learn multiple different words that map to the same image.
>
> * What is the "few shot classifier"? Yes, a multi-class classifier learns a quite different task from CoLLIE, which is precisely the point we tried to make, as well as illustrate in Figure 1. We make a comparison with such a classifier, since it can be used in a straightforward manner for continual learning. We use exactly the same few-shot classifier (based on logistic regression) as the one used in the original CLIP paper. We are not aware of any other proposed continual learning approach that can learn new mappings between embedding spaces that we can compare with, and that is why we think our approach is novel.
>
> * Comparison with no scaling: Our point here is not to make a fair comparison with CoLLIE without scaling, but to study the importance of the scaling function. A and S are two different “modules” and should not be put against each other. In fact, we first tried to use the negative examples directly for the adjustment function (with the target of outputting a zero-length vector), but this did not yield a good performance. We do not have a good explanation at this point why this was the case.
>
> * Footprint of training examples: Our use case scenario is an agent interacting with humans and an environment, continuously updating its model with continual learning. Thus, we think the footprint of the training examples needed in this process is relevant.
>
> * The problem of fine-tuning CLIP is not just catastrophic forgetting, but it is also unclear how you would be able to retrain the entire model with just one example in an efficient and effective way (i.e., to fulfil the objectives set out in the introduction of the paper).
>
> * It is true that the similarity of the transformed text embedding might be far from the target image embedding, but that is actually true in general for CLIP text/image embeddings (the dot product is not very close to 1, even if they are semantically close to each other). Unfortunately, we do not have a method for decoding images from those transformed embeddings, even if that would have been interesting.
>
> * It is impossible to get the learning curve of CoLLIE, as seen in Figure 6, if it was not able to make use of compositional semantics, since there are 85 unique color/shape combinations. The comparison with the few-shot learner in the same figure (which does not have this compositionality) makes this clear.

---

> > ### Comment · Reviewer_jzf6 · 2021-11-22
> > **Response to author comments**
> >
> > Thank you for taking the time to give responses.
> >
> > > However, we are not aware of any previous work that learns a direct transformation of the representations themselves to accomplish both continual learning and retained zero-shot use of those representations with the same model.
> > - But any model that passes the CLIP embeddings through a linear layer with bias can also be said to "learn a direct transformation of the representations"
> >
> > > We are not sure we understand what you mean by scaling the approach to “multiple tasks”?
> > - As typically described, continual learning entails the ability to "learn and remember multiple tasks from dynamic data distributions" (for example see: https://www.sciencedirect.com/science/article/pii/S0893608019300231)
> > So I believe that showing how the method scales to multiple tasks is critical if the system can be said to solve continual learning. For example, a multiple-task setup could include a tangram description task as well as a scene description task and a style description task. The point is to show that multiple domains can be accommodated without compromising performance on any of them. If this cannot be done for more than one task at a time, then I do not think it meets the criteria for "continual learning".
> >
> > > A and S are two different “modules” and should not be put against each other. In fact, we first tried to use the negative examples directly for the adjustment function (with the target of outputting a zero-length vector), but this did not yield a good performance.
> > - If the point of the ablation test is to show the effectiveness of a certain module, then ideally the training should be fixed while the presence of the module is varied. For this reason, the training examples that A and S see should be as similar as possible. However, your second sentence answers my question in this respect. Thank you.

---

> > > ### Author Response · Authors · 2021-11-22
> > > **Response to reviewer**
> > >
> > > Thanks again for your response. Here are our answers.
> > >
> > > > But any model that passes the CLIP embeddings through a linear layer with bias can also be said to "learn a direct transformation of the representations"
> > >
> > > Yes, if you only take the first part of our novelty claim, it doesn’t look very novel. But the full quote reads: “we are not aware of any previous work that learns a direct transformation of the representations themselves **to accomplish both continual learning and retained zero-shot use of those representations with the same model**”
> > >
> > > > As typically described, continual learning entails the ability to "learn and remember multiple tasks from dynamic data distributions"
> > >
> > > Yes, but the question is what is regarded as a “task”. Shin et al. (2017) write “The most common experimental formulation used in continual learning literature [34, 18] is a simple image classification problem where the inputs are images from MNIST handwritten digit database [19], but pixel values of inputs are shuffled by a random permutation sequence unique to each task. The solver has to classify permuted inputs into the original classes.” Thus, a new “task” here is just that the input distribution changes, but the problem (digit classification) is the same. Another new “task” that is often added is simply to add classes to an image classification problem. Thus, they are not necessarily going from something like “scene description” to “style description”, as you suggest. In our tangram experiment, we add a new “class” (i.e. a new combination of shape/color) in almost every round (in the beginning), and we see how well they are remembered when they are eventually picked again, or when a similar tangram is picked, making sure they do not interfere with each other. In the LAD experiment, we learn new words, while checking that the original “task” of associating words with images (that CLIP was trained for) is retained.
> > >
> > > Parisi et al. (that you quote), also write: "A lifelong learning system is defined as an adaptive algorithm capable of learning from a continuous stream of information, with such information becoming progressively available over time and where the number of tasks to be learned (e.g., membership classes in a classification task) are not predefined. Critically, the accommodation of new information should occur without catastrophic forgetting or interference." Thus, they also exemplify “task” with "membership class" in a “classification task”, and we think that the experiments we have performed are good examples of continual learning in action. However, compared to previous work in this area, our problem formulation is a bit atypical (and novel), as we don’t have a limited set of classes to begin with, since we are building on top of the zero-shot approach to “classification” used in CLIP and similar models (as illustrated in Figure 1).
> > >
> > > > However, your second sentence answers my question in this respect. Thank you.
> > >
> > > As you might have seen, we re-ran those experiments and added these results to the new version of the paper. (See the general comments related to that)

---

> > > > ### Comment · Reviewer_jzf6 · 2021-11-29
> > > > **Response to author comments**
> > > >
> > > > I thank the authors for their clear and thorough response. However, my recommendation remains the same: marginally below the acceptance threshold.
> > > >
> > > > While the work is written clearly and motivated well, I am not convinced of its technical novelty or impact. My main objections remain rooted in (1) the existence of similar approaches (mentioned above) and (2) lack of evidence that this approach can scale to multiple tasks (see discussion below).
> > > >
> > > > > Thus, they also exemplify “task” with "membership class" in a “classification task”, and we think that the experiments we have performed are good examples of continual learning in action.
> > > >
> > > > I disagree with this interpretation of Parisi et al. 2019. Lifelong learning entails the learning of multiple "tasks", where each task should come from its own distinct domain. I believe that this is what Parisi et al. meant when referring to "membership classes". As the authors note, there can be some leeway in defining what "distinct" means, but in this case, I think it is clear that there are really only two tasks that CoLLIE needs to learn: the source CLIP task and the tangram task. I recommend that the authors consider re-working/extending their CoLLIE approach so that it can handle a variety of disparate domains.

---

> > > > > ### Author Response · Authors · 2021-12-06
> > > > > **Response to reviewer**
> > > > >
> > > > > We would like to thank the reviewer again for the answer, but also comment on the two novelty issues that are left:
> > > > >
> > > > > * We do acknowledge that the method bears some similarities to GLU and residual connections in its use of a scaling function (similar to a gating function). However, we think the mechanism is applied very differently here, on top of the model’s embeddings and trained separately, rather than trained together with the rest of the deep neural network. We don’t think that once a general concept such as gating functions has been used once, no more papers using a similar mechanism (in whatever way) can be considered novel, especially when considering both technical and empirical novelty.
> > > > > * Concerning the term “continual learning”, we do not agree with the reading of Parisi et al. In section 3.5 (Benchmarks and Evaluation Metrics), “incremental class learning” is mentioned as an example of continual learning, where “the model performance reflects its ability to retain previously learned information while incrementally learning one class at a time.”. Also, the reviewer does not comment on our quote by Shin et al. (2017), a paper published at NeurIPS, which clearly suggests an understanding of the term that is in line with ours.

---

### Official Review · Reviewer_Y4pM · 2021-11-04

**Correctness:** 4
**Technical Novelty And Significance:** 3
**Empirical Novelty And Significance:** 3
**Recommendation:** 6
**Confidence:** 4

**Main Review:**

The method is simple but effective for the setting it was evaluated in. If the “Related Work” section is accurate, it is introducing a novel and interesting problem setting. The problem itself is well motivated.

My main concerns are the following:

C1) The method experiences catastrophic forgetting, as the number of classes increases (as shown in Table 1). The authors propose to mitigate it by keeping more negative examples, however this is not desirable for a continual learning setting.

C2) While the authors make an argument that this is a continual method, there are only 2 tasks at any given point - the source task that the original model was pre-trained on, and the new task, consisting of all the classes that need to be classified. Also, it appears that the solution to this setting is based on MTL - having examples from all tasks (examples from the source task are used as negative examples for the gating function) and training the free parameters on them simultaneously. As a result, I’m hesitant to support this being called a continual learning solution.

Note: there's a typo in section 3 using MMR instead of MRR.

Overall, it appears that this paper presents a simple method in an interesting direction, and can server as a useful baseline for further research.

**Summary Of The Paper:**

The paper augments a pre-trained multi-modal model, which encode language and images into a shared latent space. Said model needs adjustments in order to account for previously unseen links between phrases and images. The paper adjusts this by learning a linear transformation of the embedding space. A threshold function is used in order to apply the new linear transformation only to relevant inputs.

**Summary Of The Review:**

Overall, it appears that this paper presents a simple method in an interesting direction, and can server as a useful baseline for further research.

---

> ### Author Response · Authors · 2021-11-10
> **Replies to your concerns**
>
> Thank you for your helpful review. We are glad you found the problem setting and motivation interesting and well motivated. Regarding your main concerns:
>
> C1) We do not think the decrease in performance should be characterized as catastrophic, even if it declines when many classes are used. Some degree of forgetting should be expected, given the simplicity of the model. We think that it is indeed feasible to keep more negative examples, as these are fixed and not dependent on the specific scenario. As we mention, it is likely possible to sample these from a generative model (similar to Shin et al. 2017), which would be an interesting topic for future work.
>
> C2) If the “source task” refers to the original task CLIP was trained on (a parallel corpus of 400 million images with their captions with a contrastive learning objective), the negative examples we use (generic nouns from the English language) to train the scaling function are indeed part of CLIP’s latent space. But we do not keep language/image pairs from this training data and continue to train on these mappings. One of the novel features of CoLLIE is that we do not need to do this, since we are only predicting the difference vector that is needed for the adjustment. We therefore do not think that it is fair to say that we continue to train on the source task. Furthermore, previous work we refer to (e.g. Shin et al. 2017), do use source task data (in their case sampled from a generative model) to avoid catastrophic forgetting, and still refer to it as continual learning.

---

### Author Response · Authors · 2021-11-18
**Comments to new version of the paper**

We have now submitted a new version with the following notable changes, to address some concerns by the reviewers:

* We have added an acknowledgement that the method of using an adjustment and a scaling function has similarities to GLU and residual neural networks (even though these mechanisms are used for a different problem).

* We have added another model to Experiment I, where no scaling function is used, but where the adjustment function is also trained with the negative examples (the common nouns), to predict zero adjustment. This shows that a separate scaling function gives a much better performance. (See Figure 4).

* We have added experiments with different implementations of the scaling function, showing that they have different strengths, but that SVR has an arguably better overall performance. (See Table 2 in the Appendix).

* Two of the reviewers were not entirely convinced that Experiment II showed that CoLLIE benefits from compositional semantics, and that we should have used a “held-out test-set of unseen color+shape combinations”. To further strengthen this claim, we have therefore added a follow-up experiment, where we first train the model on all 17 shapes of one random color, and then evaluate it on the same shapes with different random colors. This was iterated 100 times. Whereas the CLIP baseline model (and the few-shot learner, which has to fall back on the CLIP model) only had an average MRR of 0.317 on these unseen combinations, CoLLIE achieved an MRR of 0.857.

---

### Decision · Program_Chairs · 2022-01-20

**Decision:**

Reject

**Comment:**

This manuscript presents a method built on top of CLIP which transforms language embeddings to understand new phrases while maintaining the original abilities of CLIP.

1. The technical novelty of this work is limited. That being said, if the task is of wide interest, a straightforward approach that performs well is not just good, it's even preferable to one that is complex. The authors bring up the fact that reviewers are asked to rate both technical and empirical novelty. This is true. Yet, reviewers were unconvinced both by the method and the setting.

  The manuscript does not explain why this setting provides additional challenges or value compared to the many other continual learning or zero-shot settings that exist in the literature. What the authors say "we are not aware of any previous work that learns a direct transformation of the representations themselves to accomplish both continual learning and retained zero-shot use of those representations with the same model" seems to be undisputed by the reviewers. But is it critically important to future ML research that a single model does both? Or that a model learns a direct transformation to do so? Overall, this task seems very constrained and tailored to this one approach, while usually the more general a setting is, the more it is valued by the community because it will be more likely to stand the test of time and lead to new advances. Reviewers also could not point to a compelling immediate practical need for such a model, which would be another reason for considering a novel setting.

  While in the responses the authors acknowledge that they do not consider their method to be the ultimate solution, that the method has significant limitations, and that this really should be considered a strong baseline, this is not how the work is presented. Relatively little is said in the manuscript about any of these topics.

2. In response to requests for experiments (such as exploring the space of transformations and exploring alternate models to CLIP) the authors put forward that space limitations preclude such experiments. I would encourage authors not to rely on this argument going forward as it does not serve their cause well. Between the unlimited appendix and the possibility of linking to an anonymized website space cannot be a constraint. Science is complex these days and it's not unusual to have to report extensive additional experiments outside the main body of the manuscript. I encourage authors to consider that these requests by multiple reviewers are likely going to be the first questions that the readers of their work will also want answers to. Exploring other transformations and models is critical to understanding the value and impact of the work.

Minor point: I did not see this in the reviews but Figure 1(a) has the labels for CLIP text and image flipped.

If the authors round out the experiments and demonstrate either that their idea is more general-purpose, i.e., that it can be applied to other settings and problems, or that this setting is of great value on its own, this could be a strong contribution.

---

> ### Public Comment · ~Gabriel_Skantze1 · 2022-02-15
> **Authors' comments on paper decision**
>
> We regret to hear that the paper was rejected, while we thank the reviewers and the program chairs for their time. There are still a few points in the decision letter that we would like to address:
> * The PCs accept our novelty statement that there is no "previous work that learns a direct transformation of the representations themselves to accomplish both continual learning and retained zero-shot use of those representations with the same model". At the same time, they ask whether it is "critically important to future ML research that a single model does both". That, to us, is a rather surprising question. It is a bit like asking why the same model should recognize both apples and oranges: why can't you have two separate models for those? We think it is indeed favorable, if not critical, that the same model can both recognize all the regular language use, while at the same time also recognize the newly learned language; how would you otherwise know which model to apply when? Our proposed method of learning a direct transformation of the representations is one way to accomplish this.
> * A main argument against the paper seems to be a lack of motivation for why this kind of model is needed and why "this setting is of great value on its own". This is also surprising for us to read, as we tried quite hard in the introduction of the paper to explain that the ability of humans to adapt their language use to new domains and to new conversation partners is fundamental, and that this is also essential for artificial agents communicating with humans. The PCs also say that the "reviewers were unconvinced both by the method and the setting" and that "Reviewers also could not point to a compelling immediate practical need for such a model". However, the first reviewer wrote that we are "introducing a novel and interesting problem setting. The problem itself is well motivated". The second reviewer wrote "Clear and thoughtful discussion of the motivation, namely the importance of grounded language learning and continual learning. I enjoyed reading it!". Thus, we do not think the reviewers' concern had to do with the lack of motivation for the problem we had formulated.
> * Other conferences we have submitted to seem to have a more restrictive view on appendices, which should not really include primary results, but rather be used for examples, etc. We now understand better ICLR’s standpoint on page limitation, i.e., that page limit is not really an issue, and that primary results can (and should) be included in appendices, as well as external websites ("Between the unlimited appendix and the possibility of linking to an anonymized website space cannot be a constraint"). At the same time, we note that the instructions in the ICLR CFP states that reviewers are not required to read appendices. Thus, this must mean that authors are required to provide material that the reviewers are not required to read. We are not sure whether it is good practice for material that is considered essential (and/or on which an acceptance decision can thus be based) to be relegated to the appendix or an external platform, especially when it is optional reading for reviewers. However, we thank the PCs for this information and will take it with us, should we consider to submit a paper to ICLR in the future.